# Insights into the Identification of the Specific Spoilage Organisms in Chicken Meat

**DOI:** 10.3390/foods9020225

**Published:** 2020-02-20

**Authors:** Cinthia E. Saenz-García, Pilar Castañeda-Serrano, Edmundo M. Mercado Silva, Christine Z. Alvarado, Gerardo M. Nava

**Affiliations:** 1Departamento de Investigación y Posgrado de Alimentos, Universidad Autónoma de Querétaro, Querétaro 76010, QRO, Mexico; elizabeth.saenz.garcia@gmail.com (C.E.S.-G.); mercado501120@gmail.com (E.M.M.S.); 2Facultad de Medicina Veterinaria y Zootecnia, Universidad Nacional Autónoma de México, Tláhuac 13300, CDMX, Mexico; pilarcs@unam.mx; 3Department of Poultry Science, Texas A&M University, College Station, TX 77843, USA; calvarado@poultry.tamu.edu

**Keywords:** meat spoilage, *Pseudomonadaceae*, genomic, specific spoilage organisms, chicken

## Abstract

Poultry meat deterioration is caused by environmental conditions, as well as proliferation of different bacterial groups, and their interactions. It has been proposed that meat spoilage involves two bacterial groups: one group that initiates the deterioration process, known as specific spoilage organisms (SSOs), and the other known as spoilage associated organisms (SAOs) which represents all bacteria groups recovered from meat samples before, during, and after the spoilage process. Numerous studies have characterized the diversity of chicken meat SAOs; nonetheless, the identification of the SSOs remains a long-standing question. Based on recent genomic studies, it is suggested that the SSOs should possess an extensive genome size to survive and proliferate in raw meat, a cold, complex, and hostile environment. To evaluate this hypothesis, we performed comparative genomic analyses in members of the meat microbiota to identify microorganisms with extensive genome size and ability to cause chicken meat spoilage. Our studies show that members of the *Pseudomonadaceae* family have evolved numerous biological features such as large genomic size, slow-growing potential, low *16S rRNA* copy number, psychrotrophic, and oligotrophic metabolism to initiate the spoilage of poultry meat. Moreover, inoculation experiments corroborated that these biological traits are associated with the potential to cause chicken meat deterioration. Together, these results provide new insights into the identification of SSO. Further studies are in progress to elucidate the impact of the SSO on meat quality and microbiota diversity.

## 1. Introduction

Microbiological quality of chicken meat is affected by handling during catching, transport, processing, and storage [1]. After these steps occur, chicken meat holds a diversity of bacterial groups known as microbiota [2]. Proliferation and interactions between members of the microbiota leads to deterioration of poultry meat [3]. For decades, numerous efforts have been made to identify bacterial groups responsible for meat spoilage; as a result, it has been proposed that meat microbiota is the result of microbial associations between two important bacterial groups: spoilage associated organisms (SAOs) and specific spoilage organisms (SSOs) [4].

The SAOs represent all bacterial genera recovered from meat samples before, during, and after the spoilage process. This group includes members of the *Pseudomonadaceae*, *Listeriaceae*, *Enterobacteriaceae, Staphylococcaceae, Shewanellaceae, Moraxellaceae, Carnobacteriaceae, Aeromonadaceae*, and *Leuconostocaceae* families [4,5,6]. In contrast, the SSO group represents a fraction of the SAO population and it is considered the direct cause of food spoilage [4,7,8]. Many biological traits remain unknown; nonetheless, it has been show that SSOs have a lower growth rate compared to SAO members [9,10]. Also antagonistic (e.g., competition for nutrients) or cooperative interactions (e.g., cross-feeding) between SSOs and SAOs have been described [7]. Isolation and identification of the SSO has been challenging due the complex interactions among members of the meat microbiota and the lack of adequate microbiological methods. For instance, analysis of meat microbiota by conventional methods cannot differentiate between bacterial groups that initiate meat spoilage and bacterial populations altered by this process [9]. Moreover, molecular studies have revealed that some bacterial species with spoilage potential cannot be isolated in reference media [10]; thus, there is a need to develop new microbiological strategies to identify bacteria belonging to the SSO group.

It has been proposed that the SSOs initiate proliferation after glucose is consumed by fast-growing members of the SAO group [11,12,13,14,15,16], suggesting that the SSOs have evolved different metabolic features to obtain nutrients from poultry meat. To gain insights into the identification of the SSOs, it is important to take advantage of the recent progress in microbial genomics to uncover some of the SSO biological traits. For example, it has been shown that bacteria adapted to cold and hostile environments possess an extensive genome size [10]. Also, bacteria with large genomes have evolved regulatory networks to proliferate in complex sites were nutrients are scarce or not readily available [17,18]. Based on these premises, the working hypothesis of the present study was that members of the poultry meat microbiota with extensive genome size, slow-growing phenotype, and ability to cause chicken meat spoilage could represent the SSOs of poultry meat. Therefore, comparative genomic analyses and in vitro experiments were performed to corroborate this idea.

## 2. Materials and Methods

### 2.1. Genomic Analyses of Bacterial Groups Associated with Spoilage in Chicken Meat

A comprehensive list of bacterial groups linked to chicken meat spoilage was integrated by performing an exhaustive literature review and analysis. Three renowned databases (PubMed, Google Scholar, and Science Direct) were used to find and retrieve peer review articles describing microbial analysis on chicken meat. Keywords such as Bacteria, Chicken, Isolation, Meat, Microbiota, *Pseudomonas*, and Spoilage were used for the searches. Bacterial genera reported in the literature were recorded and classified at the family level, using the Hierarchy Browser tool at the Ribosomal Database Project [19]. The list of bacterial families associated to chicken meat spoilage was used to retrieve whole genome information from the Integrated Microbial Genomes and Microbiomes (IMG/M) system [20]. Genomes were annotated by the DOE-JGI Microbial Genome Annotation Pipeline [20], and information such as genome size, number of protein-coding genes, and *tRNA*s was compared between different bacterial families.

### 2.2. Isolation of SSO Candidates from Spoiled Chicken Meat

To isolate potential SSOs, a novel microbiological strategy was designed to isolate bacteria adapted to cold and hostile environments. To accomplish this objective, 12 fresh skinless chicken breasts obtained from a poultry processing plant were aseptically transferred into sterile plastic containers and stored aerobically during 14 days at 4 °C. At days 10 (*n* = 10) and 14 (*n* = 10), subsamples (25 g) of meat were rinsed with 50 mL of sterile saline solution (SS), manually massaged for 3 min, and serially diluted (1:10) in SS. Dilutions were plated on three different oligotrophic agars, (1) 1.7 g tryptone, 0.05 g NaCl, 0.25 g K_2_HPO_4_, and 0.25 g glucose per liter of distilled water; (2) 0.17 g tryptone, 0.005 g NaCl, 0.025 g K_2_HPO_4_, and 0.025 g glucose per liter of distilled water; and (3) 0.0017 g tryptone, 0.0005 g NaCl, 0.0025 g K_2_HPO_4_, and 0.0025 g glucose per liter of distilled water. Bacteriological agar (16 g/L of distilled water) was added to all media. Agar plates were then incubated at 4 °C for 96 h. Representative colonies (*n* = 15–17) with different morphology were selected from each medium and pure cultures were obtained. Each isolated bacterium was subjected to DNA extraction using a commercial kit (Quick-gDNA Zymo Research, USA). Genomic DNA was used for near full-length (~1500 nt) *16S rRNA* gene PCR amplification with primers 8F (AGAGTTTGATCCTGGCTCAG) and 1510R (CGGTTACCTTGTTACGACTT [21]. The PCR protocol consisted of an initial step of 3 min at 94 °C and 35 cycles of 45 s at 94 °C, 30 s at 55.3 °C, and 30 s at 72 °C, followed by a final extension of 5 min at 72 °C. PCR products were purified and subjected to Sanger sequencing using an ABI 3730XL capillary sequencer. Inspection, alignment, and trimming of sequences were performed with MEGA6 software [22]. Sequences were classified using Seqmatch (RDP Release 11.5) tool at the Ribosomal Database Project [19].

### 2.3. Bacterial Growth Rate Analysis

Members of five bacterial families, *Aeromonadaceae* (*Aeromonas* sp. *n* = 1), *Enterobacteriaceae* (*Hafnia* spp. *n* = 3; *Enterobacter* spp. *n* = 3), *Listeriaceace* (*Brochothrix* spp. *n* = 3), *Moraxellaceae* (*Acinetobacter* spp. *n* = 2), and *Pseudomonadaceae* (*Pseudomonas* spp. *n* = 12) were used to estimate maximum growth rates. These isolates (*n* = 24) were obtained from the laboratory stock collection and all of them were recovered from spoiled chicken meat. Cultures were grown overnight in Tryptic Soy Broth (TSB; DIBICO, México) and an aliquot (200 uL) were individually placed into 25 mL of fresh TSB and incubated at 22 °C, a temperature which allows growth of the different genera isolated. Subsamples (200 uL) were collected every four hours, from 0 to 60 h of incubation, to measure optical density (550 nm) using a spectrophotometer (Varioskan Flash, Thermo Scientific; Waltham, MA. USA). Two independent experiments were performed, each carried out in triplicate. Growth curves were fitted using DMFit software [23] based on the model of Baranyi and Roberts [24] to calculate the parameters of maximum growth rate (μmax).

### 2.4. Analysis of 16S rRNA Copy Number

To corroborate phenotypic differences observed during growth curve analysis, the number of *16S rRNA* copies of each bacterial family was estimated. This genetic information was retrieved from the Ribosomal RNA Operon Copy Number Database, a curated resource for copy number information for Bacteria and Archaea [25]. For each bacterial family, the number of *16S rRNA* copies was obtained using the Search Taxonomy tool and then retrieved and archived for statistical analysis.

### 2.5. Analysis of Meat Spoilage Potential

To evaluate the capacity of different bacterial isolates to cause meat spoilage (as assessed via the quantification of total volatile basic nitrogen [TVB-N, >20 mg/100 g] [5]) inoculation experiments were performed. First, fresh skinless chicken breast (~25 g) were decontaminated by rinsing each piece of meat five times with sterile distilled water, three times with a 20% chlorine solution, and one additional time with sterile distilled water. The piece of meat was then submerged into 96% ethanol and flamed to dry [26]. The burned meat surface was removed using a sterile knife. Decontaminated meat was placed into a sterile plastic container to perform inoculation experiments as described elsewhere [27]. Briefly, the decontaminated meat samples (5 pieces per group) were inoculated (~4.0 log CFU/g) with either *Pseudomonas* sp.*, Brochothrix* sp., *Hafnia* sp., *Acinetobacter* sp. or saline solution (negative control). These isolates were recovered from spoiled chicken meat as described in Section 2.2. All samples were incubated at 4 °C and subsamples for TVB-N analyses were collected at 0, 2, 4, 6, and 8 days of incubation. This experiment was performed in triplicate.

### 2.6. Quantification of TVB-N

The method of Conway microdiffusion was used for qualification of TVB-N [5]. Briefly, 5 g of meat were homogenized using 15 mL of sterile distilled water for 1 min. This homogenate was mixed with sterile distilled water to bring the volume up to 50 mL; the mixture was filtered through a Whatman paper (No. 1). One milliliter of the filtered mixture was placed into the outer compartment of the Conway unit, and 1 mL of TVB-N reagent (0.66%: methyl red and 0.33% bromocresol green in alcohol 1:1) was added into the inner compartment of the unit. Then, one milliliter of saturated K_2_CO_3_ was added to the outer compartment and the unit was closed immediately. The Conway unit was incubated at 37 °C for 2 h and then 0.02% N H_2_SO_4_ was added to the inner compartment for titration. The concentration of TVB-N was estimated as described elsewhere [28].

### 2.7. Statistical Analysis

Results were analyzed by means of ANOVA Fisher’s protected least significant difference test using StatView version 5.0.1. Differences were considered significant at *p* < 0.05.

## 3. Results and Discussion

### 3.1. Genomic Analysis of Bacteria Associated with Chicken Meat Spoilage

A comprehensive revision of literature (>25 published articles), covering a time period from 1996 to 2018, revealed that 11 bacterial families comprising 14 genera have been isolated from spoiled chicken meat samples (Table 1). To identify SSO candidates, based on large genome size and slow-growing phenotype, a series of comparative genomic analyses were performed. A total of 11,412 bacterial genomes (Table 2) were evaluated to identify three important traits: genome size, number of genes, and *tRNA* genes. These analyses revealed that genome size in SAOs varied between 6.2 and 1.8 Mb. On average, the *Pseudomonadaceae* family exhibited the largest (*p* < 0.05) chromosome (6.2 Mb) and *Leuconostocaceae* the lowest one (1.8 Mb; Figure 1A). Also, it was revealed that the number of protein coding genes varied between 5833 and 1945 in members of SAO group. On average, the *Pseudomonadaceae* family possessed the largest (*p* < 0.05) number of protein coding sequences (5833 genes) and *Leuconostocaceae* the shortest number (1945 genes; Figure 1B). These results suggest that members of the *Pseudomonadaceae* family possess an extensive genomic repertoire with potential to survive meat processing conditions, colonize muscle tissue, and extract nutrients from this complex matrix consisting of myofibers, as well as connective and adipose tissue. Therefore, *Pseudomonadaceae* species could represent one of the SSO in chicken meat. This idea is supported by numerous reports showing that bacterial adaptation to cold, complex, and hostile environments is driven by genome size and gene content [17], these traits provide bacteria with a wide metabolic versatility for a successful survival in hostile milieus such as temperature variations and chemical interventions as observed during the processing of chicken. Large genomes (>5500 genes) provide bacteria with large networks to regulate gene expression and rapid adaptation to diverse environmental conditions [18,29].

At the initial stage of meat storage, glucose, lactic acid, and water-soluble proteins are consumed by fast-growing members of the SAO group; therefore, SSOs initiate a slow proliferation in this nutrient-depleted milieu by extracting nutrients from muscle tissue [4,8].Interestingly, it has been shown that slow-growing bacteria possess a low number of *tRNA* genes in their genome [44,45]. Thus, the number of *tRNA* genes was analyzed in the 11 families of chicken meat SAOs. The analysis showed that this bacterial group possess an average and median of 64 and 56 *tRNA* genes, respectively, in their genomes [44,45]. The largest number (93 genes) was observed in the *Aeromonadaceae* family whereas the lowest number was found in *Leuconostocaceae* (46 genes). Members of the *Pseudomonadaceae* (56 genes), *Staplylococcaceae* (56 genes), *Listeriaceae* (54 genes), *Carnobacteriaceae* (53 genes), and *Enterococcaceae* (52 genes) families possess the second lowest number of *tRNA* genes (Figure 1C), suggesting that these bacterial groups possess a slow-growth rate. It has been reported that fast-growing bacteria require a large number of *tRNA* gene copies for rapid protein translation and multiplication. In contrast, bacteria where speed of translation is not a limiting factor, copies of *tRNA* genes are reduced [44,45,46]. Also, it has been proposed that a low number *tRNA* is an biological adaptation to success in environments with limited nutrient concentrations [46]. Taken together, these results suggests that, under these experimental conditions, members of the *Pseudomonadaceae* family are the most predominant group possessing an extensive genome encoding for a versatile collection of metabolic enzymes and low number of *tRNA* genes for a slow-growth in a hostile environment.

### 3.2. Isolation of SSO Candidates from Spoiled Chicken Meat

It has been reported that conventional agar media cannot allow the recovery of bacteria with a potential to cause meat spoilage [4]; thus, a novel microbiological strategy was developed. Our working hypothesis was that an extensive genome size and low *tRNA* copy number allow SSOs to proliferate under showed environments. Therefore, to address this goal, it is important to design bacterial culture media that are able to select psychrotrophic bacteria with large genomes. Extensive genome size allows microorganisms to succeed in cold and oligotrophic environments [47,48,49]. Based on this information, an oligotrophic medium was designed and used to recover psychrotrophic and slow-growing bacteria (~96 h incubation). Using this approach, a total of 47 bacterial isolates were recovered from spoiled chicken meat. Amplification and sequencing of the *16S rRNA* gene revealed that the majority of the isolates belonged to genus *Pseudomonas* (43/47, 91.5%), follow by the genera *Brochothrix* (2/47, 4.3%), *Hafnia* (1/47, 2.1%), and *Acinetobacter* (1/47, 2.1%). These genera belonged to four families (*Pseudomonadaceae*, *Listeriaceae, Enterobacteriaceae*, and *Moraxellaceae*, respectively). These results indicate that members of the *Pseudomonadaceae* family are the most predominant group able to proliferate under stressful conditions such as psychrotrophic and oligotrophic environments, as found in meat processing conditions, and could represent one of the SSOs in chicken meat.

### 3.3. Identification of Slow-Growing Bacteria

To corroborate the slow-growing phenotype of these SSO candidates, maximum growth rates were estimated using twenty-four isolates recovered from chicken meat. These isolates belonged to five different bacterial families. As expected, members of *Pseudomonadaceae* family showed the lowest maximum growth rate (*p* < 0.05), followed by members of the *Moraxellaceae*, *Enterobacteriaceae Aeromonadaceae,* and *Listeriaceae* families after 60 h of incubation (Figure 2A). Interestingly, maximum growth rate correlated (R = 0.688; *p* < 0.05) negatively with bacterial genome size (Figure 2B). Also, differences in maximum growth rates were associated with lower *16S rRNA* copy numbers showing that members of *Pseudomonadaceae* (4.9 genes) possess the lowest (*p* < 0.05), number of copies followed by *Listeriaceae* (average = 5.7 genes), *Moraxellaceae* (5.8 genes), and *Enterobacteriaceae* (7.2 genes; Figure 2C). It has been documented that low *16S rRNA* copy number represents a bacterial ecological strategy to proliferate successfully in environments with low nutrient concentrations; this molecular strategy reduces metabolic expenses because the production of ribosomes is restricted [44,50]. Taken together, these results indicate that members of the *Pseudomonadaceae* family have evolved numerous ecological features to proliferate in a complex matrix comprising a cold and hostile environment. Thus, it was hypothesized that this bacterial group represents one of the SSOs in chicken meat.

### 3.4. Proof-of-Concept: Validation of SSO Potential

To validate the hypothesis that psychrotrophic, oligotrophic and slow-growing *Pseudomonadaceae* isolates could represent one of the SSOs in chicken meat, inoculation experiments were carried out. Fresh and decontaminated chicken meat samples were inoculated with representative isolates of *Pseudomonadaceae*, *Listeriaceae*, *Enterobacteriaceae*, and *Moraxellaceae* families, and stored under commercial conditions (4 °C) for a total time period of 8 days. Meat spoilage was assessed by means of TVB-N concentrations, as it is considered one of the main chemical parameters associated to microbial spoilage in chicken meat [5,30]. Remarkably, *Pseudomonas* sp. induced meat spoilage (≥20 mg TVB-N/100g [5]) just after 4 days of storage (Figure 3). Production of TVB-N initiated on day 2 (10.4 mg TVB-N/100g) and increased gradually to reach the highest (*p* < 0.05) concentrations (40.0 mg TVB-N/100g) on day 6 of storage. Inoculation with *Brochothrix* sp. and *Hafnia* sp. caused meat spoilage on day 6 of storage; however, TVB-N production reached a plateau phase after this point. In contrast, *Acinetobacter* sp. did not generated meat spoilage as determined by TVB-N levels (Figure 3). These results represent a proof-of-concept indicating that members of the *Pseudomonadaceae* family could be considered one of the SSOs in chicken meat. This idea is supported by an independent study showing that *Pseudomonas fragi or fluorecens* are the main spoilage microorganisms when inoculated in fresh meat [30]. Also, numerous studies have indirectly linked the increase in *Pseudomonadaceae* density during storage to meat deterioration, suggesting that colonization and proliferation by this bacterial group could be the main cause of poultry meat spoilage under aerobic conditions [30,37,39,40].

## 4. Conclusions

The present study provides new insights into the identification of bacteria responsible for meat spoilage. Supported by comparative genomic analyses, we identified that SSOs have evolved numerous ecological features to proliferate in a cold and hostile environment, such as poultry meat. These characteristics involve acquisition of an extensive genome size directed to successfully survive in a psychrotrophic and oligotrophic environment; traits found in members of the *Pseudomonadaceae* family. Further studies need to be implemented to elucidate the impact of these potential SSOs on meat quality and microbiota diversity. This information is essential to establish new strategies to reduce meat spoilage.

## Figures and Tables

**Figure 1 foods-09-00225-f001:**
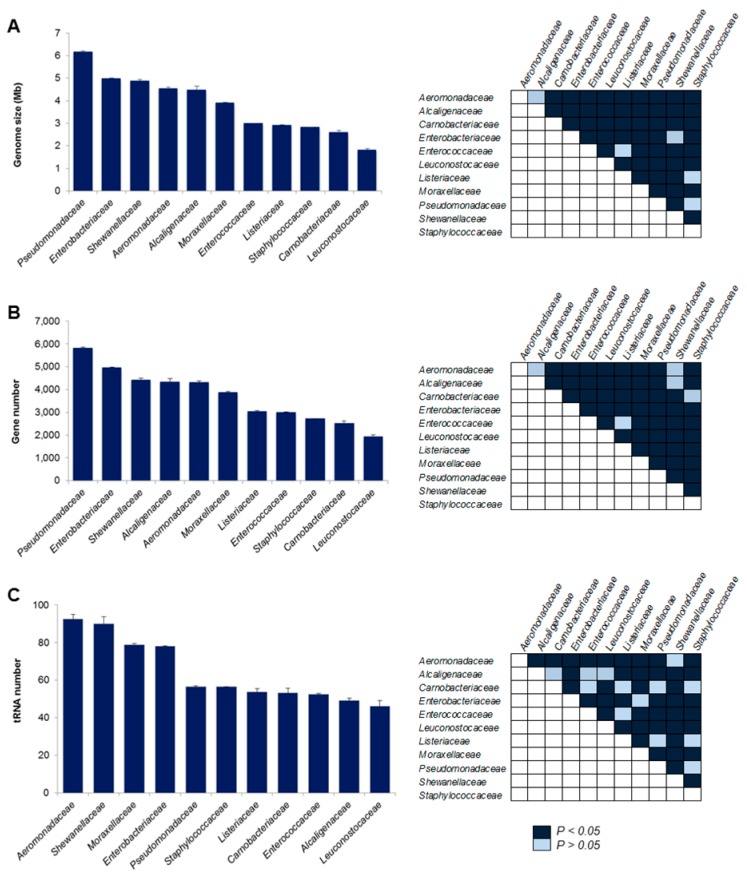
Genomic analyses of bacterial groups associated with spoilage in chicken meat. A total of 11,412 bacterial genomes comprising 11 bacterial families were analyzed to estimate (**A**) genome size, (**B**) number of genes, and (**C**) number of *tRNA* genes. (*Left*) Graphs depict mean values and its standard errors. (*Right*) Pairwise matrix showing *p* values estimated by means of ANOVA Fisher’s protected least significant difference test.

**Figure 2 foods-09-00225-f002:**
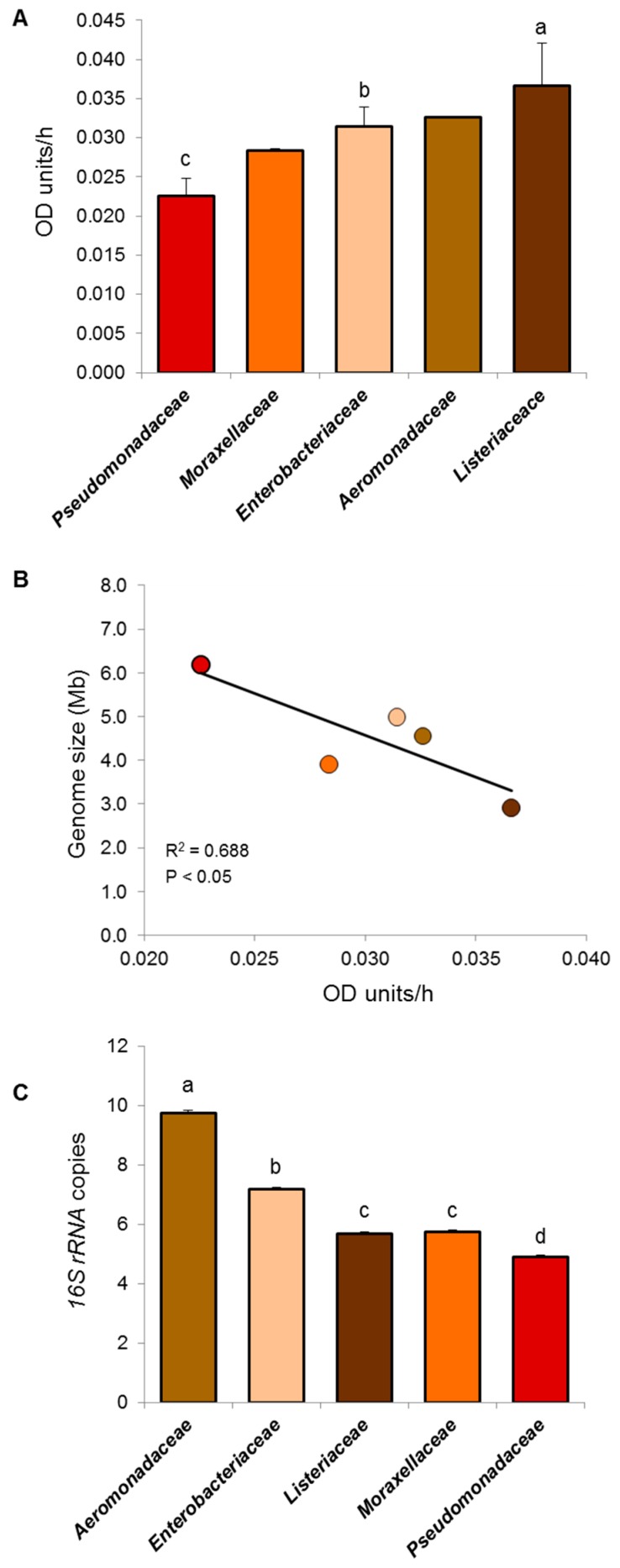
Growth phenotype and genotype of candidate Specific Spoilage Organism (SSO) obtained from spoiled chicken meat. (**A**) Maximum growth rates estimated with DMFit software and Baranyi and Roberts model. Two independent experiments were performed, each carried out in triplicate. Statistical assessment was performed only for bacterial families with >3 isolates. (**B**) Pearson correlation coefficient (R^2^) between genome size and maximum growth rate. Circle color coding as in panels A and C. OD: optical density. (**C**) *16S rRNA* copy number. Graphs depict mean values and its standard errors. Different letters are significantly different (*p* < 0.05). Statistical analysis was performed by means of ANOVA Fisher’s protected least significant difference test.

**Figure 3 foods-09-00225-f003:**
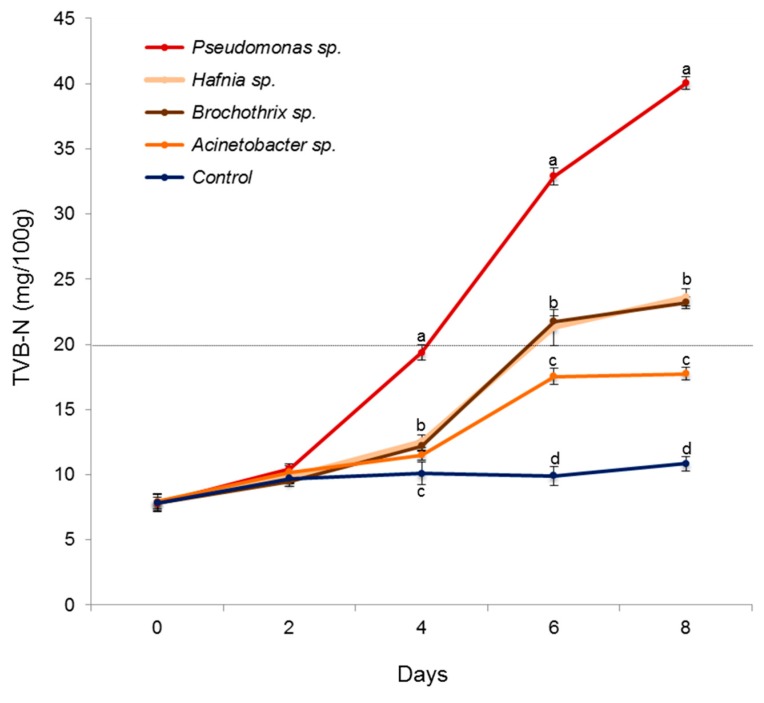
Spoilage potential of SSO candidates obtained from spoiled chicken meat. Fresh decontaminated meat samples were inoculated (~4.0 log CFU/g) with either *Pseudomonas* sp.*, Brochothrix* sp., *Hafnia* sp., *Acinetobacter* sp., or sterile saline solution (negative control) and stored at 4 °C across 8 days. Total volatile basic nitrogen (TVB-N) >20 mg/100 g (pointed gray line) was used as an indicator of meat spoilage. Graph depicts mean values and their standard errors from three independent experiments. Different letters are significantly different (*p* < 0.05). Statistical analysis was performed by means of ANOVA Fisher’s protected least significant difference test.

**Table 1 foods-09-00225-t001:** List of bacterial groups associated to chicken meat spoilage.

Genus	Family	Reference *
*Aeromonas*	*Aeromonadaceae*	[1,5,30]
*Alcaligenes*	*Alcaligenaceae*	[1,5,31]
*Carnobacterium*	*Carnobacteriaceae*	[5,31,32]
*Leclercia* *Hafnia*	*Enterobacteriaceae*	[1,5,33,34]
*Enterococcus*	*Enterococcaceae*	[1,5,31,34,35,36]
*Brochothrix*	*Listeriaceae*	[1,5,32,37]
*Leuconostoc* *Weisella*	*Leuconostocaceae*	[1,5]
*Psychrobacter* *Acinetobacter*	*Moraxellaceae*	[1,5,31,35]
*Pseudomonas*	*Pseudomonadaceae*	[1,5,9,27,30,31,32,37,38,39,40,41]
*Shewanella*	*Shewanellaceae*	[1,5,32,42,43]
*Staphylococcu* *s*	*Staphylococcaceae*	[5,9,33,42]

* A comprehensive revision of literature was performed using three renowned databases (PubMed, Google Scholar, and Science Direct), covering a time period from 1996 to 2018.

**Table 2 foods-09-00225-t002:** Number of genomes analyzed for each bacterial family associated to chicken meat spoilage.

Family	Number of Genomes
*Aeromonadaceae*	58
*Alcaligenaceae*	91
*Carnobacteriaceae*	42
*Enterobacteriaceae*	3962
*Enterococcaceae*	714
*Leuconostocaceae*	67
*Listeriaceae*	116
*Moraxellaceae*	1044
*Pseudomonadaceae*	855
*Shewanellaceae*	44
*Staphylococcaceae*	4419
Total	11,412

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
