# Peer review of "Insights into the Identification of the Specific Spoilage Organisms in Chicken Meat"

_foods, 2020, doi:10.3390/foods9020225_

Round 1
Reviewer 1 Report
The present study reports on the identification and characterization of the specific spoilage organisms in chicken meta as provided by comparative genomic analyses data. According to the findings of this study, bacterial members of the Pseudomonadaceae via various biological strategies and traits. including among others their oligotrophic and psychrotrophic character, could possess the metabolic versatility to represent the SSO in chicken meat. Given the well established association of Pseudomonas spp. with poultry meat spoilage under aerobic conditions, the findings of this study are not particularly novel. Nonetheless, the applied genomic analysis allowing for the elucidation at the genotypic level of such association and for the determination of the specific pertinent traits of these organisms in their role as SSO, make the contribution of this work to the currently available scientific knowledge valuable.
Some more specific comments, most of them minor in nature are the following:
I would recommend changing "organism" to "organisms" in the title of the manuscript. Since the findings refer to family level (and not species level), plural form seems more appropriate.
The definition of SAO and SSO as provided in the abstract of the manuscript (L15-18) is not accurate and contradicts the definition provided in L41-45. Please revise as needed.
L36: please correct "between" to "among". Similarly in L50.
L39: the SAO are also known in the scientific literature as "microbial association"; you may want to mention this as well.
L46: do you mean at "low initial concentrations"?
L59: please change to "genomic traits"
L64: revise to "...in members of poultry meat microbiota..."
L88-89: revise to "Bacteriological agar (16g/l of distilled water) was added to all media." L90: please correct to "...from each medium"; singular: medium, plural: media
L91: please correct to "Each isolated bacterium..."; singular: bacterium, plural: bacteria
L103: how is "One representative isolate from each family" defined? Since a random selection is made, this selected isolate cannot be regarded as representative...
L114: revise to "For each bacterial family, the number of 16S rRNA copies..."
L117: revise to "...meat spoilage (as assessed via the quantification of the Total Volatile Basic Nitrogen..."
L143: which time period these >30 published papers cover? Was a focus placed on the recent research findings? I am asking this because the literature regarding poultry meat spoilage is redundant...Also, which keywords were used in the research literature databases?
L144: the term "spoilage chicken" is not appropriate; do you mean "spoiled chicken" or "spoiling chicken"? Please, revise accordingly. Similarly in L227 (Figure 2 caption).
L158: please revise to "...and gene content, with these traits providing bacteria with a wide metabolic versatility..."
L184: revise to "...an extensive genome encoding for a versatile collection of..." Table 1 title: Please provide more detailed information regarding the provided studies (e.g., literature data bases used, time period covered etc.)
Figure 1: it would make more sense if both herein and throughout the manuscript, a single P-value was used a threshold for significance level, namely at α=0.05 as mentioned in L140. Thus, it would be better if only P<0.05 and P≥0.05 references are made for significant and non-significant differences, respectively. For instance P<0.005 is mentioned in L215 and 230, P<0.0001 in L239 and 257. Please use the defined significance threshold and be consistent.
L195: revise to "...cannot allow for recovery of bacteria..."
L203: correct to "an oligotrophic medium..."
L203: change to "slow-growing bacteria"; similarly wherever else mentioned in the manuscript
L206-207: please revise to "...belonged to the genus Pseudomonas, followed by the genera Brochothrix..."
L220-222: please revise the sentence "It has been documented...limited production of ribosomes" for grammar and syntax; you mention both "its" and "their" making hard for the reader to follow the sentence.
L225: prefer using third person and passive voice in your descriptions; e.g., "it was hypothesized" instead of "we hypothesized"
Figure 2 caption: Use "spp." instead of "sp.". Also correct to "their standard errors"; Make corresponding changes in Figure 3 caption.
L239, 240, 241: change "at day" to "on day"
L237: revise to "a total time period of 8 days"
L238-240: Spoilage was assessed based only on the TVB-N content, and no other aspect of microbiological spoilage was determined (e.g., sensory evaluation or determination of actual bacterial concentration). Maybe a pertinent comment should be included in the discussion of the manuscript.
L246: what about the naturally occuring microbiota? Are things expected to be different?
L260: revise to "poultry meat spoilage"
L263: change "acquirement" to "acquisition"
Reviewer 2 Report
In the paper entitled “Insights into the identification of the specific spoilage organism in chicken meat” the authors perform comparative genomic analyses in members of the meat microbiota to identify microorganisms with extensive genome size, slow-growing phenotype and the capability to cause chicken meat spoilage (as declared in line 63-65).
In the first part of the study the authors performed a comparative genome analysis on genomes of spoiler bacteria selected from the literature.
In the second part the authors isolated strains from chicken meat in a oligotrophic medium to be used to inoculate chicken meat to monitor the spoilage process.
In my opinion this study presents a very interesting approach to study the spoilage of food, studying the adaptation of specific taxa to form the SSO population, however the study present several weakness both in the first part and in the second part.
The starting hypothesis: the authors started with an hypothesis reported in line 60-63: bacteria adapted to cold, nutrient-limited and hostile environments possess an extensive genome size and bacteria with large genomes have evolved regulatory networks to proliferate in complex sites were nutrients are scarce or not readily available. However, as reported in the Introduction and Results (3.1) the hypothesis is presented as an hypothesis already validated by several previous studies (references 12-13-14-15, line 61-63, 26-27 line 158). Checking the references it seems to me that this hypothesis was not demonstrated. In reference 13 the large genomes seemed to be related to fluctuating environment, reference 12 is a pregenomic paper (1977) reporting spoilage data, 15 and 16 not reported adaptation to cold, and nutrient limited environment, in 27 a comparison between different bacterial lifestyle adaptation regarding genome size and 27 is referred to the number of transcription regulation factors (that might be related to genome size). Concluding I suggest to the authors to present the hypothesis as a new hypothesis that have to be demonstrated. The previous studies, most of them of great interest may be reported as the starting point to the author hypothesis formulation, enlarging this part of the introduction. Line 162 Pseudomonas aeruginosa is not a good example as spoiler bacterium: it is rarely isolated from food. Isolation from chicken meat of spoilage bacteria: why a culture independent method with definition of the community was not performed in parallel, to define during the aerobic conservation of the meat? In this way it might possible to evaluate the dynamics of the population from SAO and SSO. Bacterial growth curve analysis: in my opinion this is a very important weakness of the paper. Only one strain for each genus is not enough to report as representative of the genus. For the genus Pseudomonas e. several studies demonstrated very different spoilage potential both for proteolytic and lipolytic activity. A minim number of 10-15 strains belonging to the different species of Pseudomonas fluorescence group are requested and similar number for the other genera (Hafnia, Brochotrix and Acinetobacter)
Reviewer 3 Report
Reviewer report
The manuscript presents data on Specific Spoilage Organism (SSO) candidates in chicken meat. The authors provide details of some genome features (determination of mean genome size, tRNA copy number and gene number) of typical chicken meat spoilage-associated bacteria and phenotypic data (growth curve analysis and a re-inoculation experiment) from four isolates that they recovered from spoiled meat to provide insights into potential SSO candidates.
SSOs are the bacteria that cause unpleasant changes to meat texture, colour and smell. There are however other bacteria that can be isolated from meat (including spoiled meat), which do not cause detrimental changes to the product. The bacteria responsible for spoilage are not necessarily those belonging to the dominant population/s in the spoiled product. Therefore, to obtain proof that bacteria isolated from meat may indeed be SSOs, they need to re-inoculated into food model systems and markers for spoilage must be evaluated. Thus, this paper addresses a topic that would be of interest to readers from the microbial spoilage field, however, the data presented is not novel.
Major comments
The major problem of the manuscript is that the data presented is not new. The comparative genome analyses are summaries of information available in databases. Although the summaries themselves are useful, this is not new information and can be found by a quick database search. Furthermore, many studies, including studies dating back decades, have shown that Pseudomonas spp. are SSOs of refrigerated, aerobically-stored chicken (Spoilage Association of Chicken Leg Muscle, T.A. McMeekin, 1977, American Society for Microbiology; A Study of Bacteria Contaminating Refrigerated Cooked Chicken; Their Spoilage Potential and Possible Origin, G. Toule and O. Murphy, 1978, The Journal of Hygiene). Therefore, the phenotypic data, which was conducted with single isolates, provides no new insights into the SSOs of chicken meat.
There are also some major problems with the bacterial growth curve analysis. Firstly, this analysis was conducted with a single isolate per bacterial group (bacterial grouping was done at the family level). The authors mention that these are representative isolates of the groups. Considering the likely variability in growth rates among different genera of a bacterial family, as well as variability between species of the same genus and strain-specific differences, I don’t believe it is possible to draw any valuable conclusions for the bacterial groups based on the growth of a single isolate. Secondly, although the authors state that the growth rate of the Pseudomonasisolate was the slowest, growth rates and how these were determined, are not provided in the manuscript. For example, were the growth rates calculated from two data points lying within the linear section of the exponential phase of the growth curve according to the followingequation:
where Ci and Cj represent the two reads of OD600 values at two time points of tj and ti, respectively (https://doi.org/10.1186/s12866-018-1242-4 )?
Furthermore, although the claim is made, there doesn't appear to be much difference in the growth rates of the Hafniaand Acinetobacterspp.. The lag phase of the Hafniaisolate does however appear to be shorter, which the authors have not commented on. I would suggest using software like DMFit to fit the curves and obtain parameters such as growth rate and lag time.
Minor comments
Line 21: Raw meat, including poultry meat, is not nutrient-limited! It is rich in protein, fatty acids and vitamins (DOI: 10.5772/intechopen.77045). The authors repeatedly make this claim throughout the manuscript.
Line 23-26: The authors claims go beyond the results shown. The authors have not "revealed” this data as it is already known. Genome sizes and 16S copy numbers are available in databases and a multitude of papers have shown that there are psychrotrophic and oligotrophic members of the Pseudomonadaceae family (DOI: 10.1590/S1517-838246220130963, PMID: 4201647,DOI: 10.1128/JB.06217-1, DOI: 10.1007/978-1-4419-0032-6_3). Furthermore, Similar claims made elsewhere in the manuscript should also be adjusted to reflect what the results show (e.g. Lines 261-263). Also, I would suggest changing the word “strategies” for “features”.
Lines 45-48: It is unclear why SSOs should be found at low concentrations and have a slow growth rate. I could not find this information in the cited articles. If the storage conditions select for the SSOs, then not only are they likely to grow faster than other organisms for which the conditions are less favourable, but they could become the dominant species.
Line 52-54: I would suggest adding the word “some” before bacterial i.e. Moreover, molecular studies have revealed that some bacterial species with spoilage potential cannot be isolated with reference media.
Lines 56 and 57: What about the possibility that the SSOs preferentially utilise glucose and when glucose is exhausted they switch to metabolising other nutrients such as amino acids, thereby producing offensive by-products (Nychas et al., 1988)?
Lines 71 and 72: It is not clear to me why the authors chose to classify the bacterial groups at the family level. Considering the extensive differences between bacteria of different genera and even the marked phenotypic differences between bacteria of the same species, I think it would be much more informative to classify the bacteria based on at least their genera.
Line 74: There is no mention of how the genome sequences for retrieval were chosen. Were they chosen based on their genus or species, or perhaps sequencing status?
Line 75: Information such as non-coding RNAs and GC% may have been collected and analysed, but this information is not shown in the manuscript. I suggest removing these from the list.
Line 80: Please explain why this strategy is novel and not just adapted from other existing strategies? Also, the authors use oligotrophic media to facilitate isolation of bacteria adapted to nutrient-limited environments. It is not clear why this is needed. I assume the authors are referring to the harsh conditions found in food processing environments, but it is not clear at this point in the manuscript.
Line 83: Does n=0 refer to the number of samples analysed? Why were only 10 samples taken when there were 12 pieces of chicken? The origin of the 12 chicken breasts should be provided.
Line 101, 125 and 254: Brochothrixis incorrectly written as Brochotrix.
Lines 123-125: Please include more information about the isolates used to inoculate the meat pieces. Were the same isolates used for the growth curve analyses also used to inoculate the meat pieces?
Lines 166-167: I am concerned that the authors discuss that Pseudomonadaceae could be the SSO, as if there will only be one bacterial group that are the SSOs (see also Lines 156, 211, 225). This may however be a language issue, in which case the authors should employ the help of an English language editor.
Lines 168-170: As I mentioned above, the SSOs may also be able to utilise glucose and grow quickly and chicken meat is not a nutrient limited environment.
Lines 183-185: The results shown do not demonstrate that the genomes of Pseudomonadaceae encode a versatile collection of metabolic enzymes. This sentence needs to be more carefully worded.
Figure 1: Family names of bacteria have been cut off in the bottom graph on the left side of the figure. It is also unclear why P-values have been calculated for these analyses, as they do not provide any valuable information. Looking at the key for the matrices, does this mean that no P-values were obtained between 0.05 and 0.001?
Lines 198-201: How do “These traits” enable bacteria to grow at psychrotrophic temperatures? Many organisms with small- to medium-sized genomes can grow at low temperatures e.g. Lactic acid bacteria, Listeria monocytogenesand Brochothrix thermosphacta. Also, it is not the culture media that needs to be designed to isolate psychrotrophic bacteria, it is the incubation temperatures of the food matrix and agar plates that must be correctly chosen.
Lines 200-204: There is information redundancy in these sentences, complicating the point the authors would like to convey.
Line 208-211: The authors need to emphasize that Pseudomonaswere the most predominant group under the tested conditions.
Lines 214-216: The authors use the word “members” of the bacterial family. These analyses were conducted with a single isolate. This should be made clear.
Line 217: I suggest replacing “Differences in growth rate were confirmed by analysis of ….“ with “Differences in growth rates were associated with lower 16S rRNA copy numbers…”.
Lines 216-219: The authors have determined whether the differences in operon numbers are statistically different among the different bacteria. What useful information does this analysis provide?
Lines 227-230: The figure legend states that graphs depict mean values and standard errors from three independent experiments. I do not see any error bars for the growth curves. Do the data points of the growth curves represent mean values from three independent experiments conducted with one biological replicate? If so, I imagine the error bars would be quite large. Does graph B really show mean values and standard errors of three independent experiments, or does it show mean values calculated from the number of genomes available in the data rrn database. Please clarify this.
Lines 246-248: A quick search revealed articles that used chicken meat inoculation models to investigate spoilage (see below). This claim should be removed from the manuscript.
1) A Study of Bacteria Contaminating Refrigerated Cooked Chicken; Their Spoilage Potential and Possible Origin, G. Toule and O. Murphy, 1978, The Journal of Hygiene.
2) Spoilage Association of Chicken Leg Muscle, T.A. McMeekin, 1977, American Society for Microbiology.
Lines 248-250: The authors need to mention here that this bacterial group could be the main cause of meat spoilage “under aerobic conditions”.
Lines 256-257: Although stated, thereare no error bars in the graph.
Round 2
Reviewer 2 Report
I have not comments, The paper was modified according to my suggestions and in This forms is suitable for acceptation.
Author Response
Thank very much.
Reviewer 3 Report
The authors have addressed to my satisfaction the major concerns I had regarding the growth curve analyses, as well as all of the minor points I had noted. I believe the manuscript is overall much improved. I just have a couple of minor comments that need attention.
To avoid confusion, I suggest using the abbreviations SSOs and SAOs to denote the plurals of each (i.e. specific spoilage organisms and spoilage associated organisms) and SSO and SAO to indicate the singular forms of these terms (i.e. specific spoilage organism and spoilage associated organism).
Figure 2: I believe “OD units/h” are the correct units for the y-axis and x-axis of graphs A and B, respectively.
L 243: I think the authors mean “Maximum growth rates….” rather than “Maximum growth curves…”.
Author Response
Point-by-point Responses to Reviewer
Manuscript ID: foods-703288
The authors thank the Reviewers and Editor for the time invested in revising this manuscript. All minor issues have been fully addressed.
Reviewer #3
To avoid confusion, I suggest using the abbreviations SSOs and SAOs to denote the plurals of each (i.e. specific spoilage organisms and spoilage associated organisms) and SSO and SAO to indicate the singular forms of these terms (i.e. specific spoilage organism and spoilage associated organism).Response: The abbreviations were changed as suggested.
Figure 2: I believe “OD units/h” are the correct units for the y-axis and x-axis of graphs A and B, respectively.
Response: The units were changed as suggested.
L 243: I think the authors mean “Maximum growth rates….” rather than “Maximum growth curves…”.
Response: The word was changed as suggested.